# Distinct Bacterial Communities in São Jorge Cheese with Protected Designation of Origin (PDO)

**DOI:** 10.3390/foods12050990

**Published:** 2023-02-26

**Authors:** Márcia C. Coelho, Francisco Xavier Malcata, Célia C. G. Silva

**Affiliations:** 1School of Agrarian and Environmental Sciences, University of the Azores, 9700-042 Angra do Heroísmo, Portugal; 2LEPABE—Laboratory for Process Engineering, Environment, Biotechnology and Energy, Department of Chemical Engineering, Faculty of Engineering, University of Porto, 4200-465 Oporto, Portugal; 3ALiCE—Associate Laboratory in Chemical Engineering, Faculty of Engineering, University of Porto, Rua Dr. Roberto Frias, 4200-465 Oporto, Portugal; 4Institute of Agricultural and Environmental Research and Technology (IITAA), University of the Azores, 9700-042 Angra do Heroísmo, Portugal

**Keywords:** cheese, microbiota, lactic acid bacteria, *Leuconostoc*, fermented foods, metagenomic analysis, bacterial diversity, high throughput sequencing

## Abstract

São Jorge cheese is an iconic product of the Azores, produced from raw cow’s milk and natural whey starter (NWS). Although it is produced according to Protected Designation of Origin (PDO) specifications, the granting of the PDO label depends crucially on sensory evaluation by trained tasters. The aim of this work was to characterize the bacterial diversity of this cheese using next-generation sequencing (NGS) and to identify the specific microbiota that contributes most to its uniqueness as a PDO by distinguishing the bacterial communities of PDO and non-PDO cheeses. The NWS and curd microbiota was dominated by *Streptococcus* and *Lactococcus*, whereas *Lactobacillus* and *Leuconostoc* were also present in the core microbiota of the cheese along with these genera. Significant differences (*p* < 0.05) in bacterial community composition were found between PDO cheese and non-certified cheese; *Leuconostoc* was found to play the chief role in this regard. Certified cheeses were richer in *Leuconostoc*, *Lactobacillus* and *Enterococcus*, but had fewer *Streptococcus* (*p* < 0.05). A negative correlation was found between contaminating bacteria, e.g., *Staphylococcus* and *Acinetobacter*, and the development of PDO-associated bacteria such as *Leuconostoc*, *Lactobacillus* and *Enterococcus*. A reduction in contaminating bacteria was found to be crucial for the development of a bacterial community rich in *Leuconostoc* and *Lactobacillus*, thus justifying the PDO seal of quality. This study has helped to clearly distinguish between cheeses with and without PDO based on the composition of the bacterial community. The characterization of the NWS and the cheese microbiota can contribute to a better understanding of the microbial dynamics of this traditional PDO cheese and can help producers interested in maintaining the identity and quality of São Jorge PDO cheese.

## 1. Introduction

The microbiota of raw milk cheeses is quite complex and includes many non-starter lactic acid bacteria (LAB) strains originally derived from the milk itself or introduced by the manufacturing environment; these bacteria are important for the ripening of the cheese and the development of the expected flavor [1,2]. Interest in the functional and structural diversity of the microbiota in raw milk cheeses has increased because these cheeses have a more intense and unique flavor compared to cheeses produced from pasteurized milk (Montel et al., 2014). Several studies have attempted to describe the microbiota of traditional cheeses and the distinct stages of cheese manufacture and ripening [3,4,5,6,7,8].

Culture-dependent methods have been the preferred choice, but they are labor-intensive and inherently biased [9]. Therefore, culture-independent techniques and next-generation sequencing (NGS) technology have played a key role in recent studies on microbial communities in traditional cheeses [10,11,12,13,14,15]. In addition, NGS methods can reveal the existence of subdominant populations within the cheese microbiota that are difficult to detect using culture-dependent methods. These populations may be responsible (at least in part) for the differentiating flavors of raw milk cheeses. The interactions of the subdominant (or rare) microbiota with the dominant microbiota also likely play an important role in the development of the key flavor and aroma notes of these cheeses [16,17,18].

São Jorge cheese is a very popular Portuguese cheese produced from raw cow’s milk on the island of São Jorge in the Azores. It bears major economic and social importance on the island. This cheese exhibits a yellowish, hard or semi-hard paste and a crumbly texture. It is produced from raw cow’s milk to which a natural whey starter (NWS) is added, obtained from the whey of the previous day’s cheese-making. The NWS is added in a ratio of about 1.5–2/1000 (1.5/1000, for NWS acidity 50–65 °D; 2/1000, for NWS acidity 40–50 °D). Acidification takes place at 30 °C and is followed by cooking the curd at 35–36 °C, draining the whey, shaping the curd, and salting and pressing the curd. This is followed by the ripening process, which lasts at least 3–4 months. By the end of the ripening, São Jorge cheese has small, irregular eyes, and its flavor is characterized by strong, clean and slightly spicy notes that become more intense as it matures. Therefore, the indigenous microbiota of the raw milk and the starter culture (NWS) is important for the subsequent ripening process, as both actively contribute to the characteristic aroma and spicy flavor of the final product. However, the variability of the final product—expected in view of its being manufactured from raw milk, is often sufficient to compromise the PDO seal. In order to obtain this seal, it is not enough for the cheese to be produced by a certified cheese maker according to PDO specifications; each batch of cheese is indeed also subjected to sensory testing by a trained panel from an independent certifying body (Confraria do Queijo de São Jorge). As a result, a large percentage of cheeses (40–60%) produced according to PDO specifications will be eventually denied PDO status, yet they will still be suitable for selling at lower prices.

Although a few studies have attempted to identify and characterize the dominant bacteria in São Jorge cheese [19,20,21,22], they all relied on culture-dependent methods, unable to fully decipher the diversity associated with this type of dairy product. In addition, the growth media used in culture-dependent methods are not truly selective for differentiation among bacterial communities [23]. Therefore, complementary, more in-depth studies are needed to fully elucidate the role of the entire microbiota of this cheese. To achieve this goal, a detailed identification and characterization of the microbiota using culture-independent methods appears important for the eventual selection of tailor-made starter cultures (SLAB) and/or auxiliary cultures specifically designed to control the fermentation of this cheese, thus helping reduce variability, achieve the best sensory characteristics during ripening, and guarantee a higher percentage of PDO cheeses. The aim of this work was accordingly to apply culture-independent and NGS methods to characterize the bacterial communities in milk, NWS, curd, and final cheese under a concerted program to shed light on the dynamics of the microbiota and to rationalize the failure to receive the PDO seal on microbiological grounds.

## 2. Materials and Methods

### 2.1. Sampling Collection

Samples of milk, NWS, curd and cheese (9 months ripening) from the traditional production of São Jorge cheese were taken aseptically on four different occasions at “UNIQUEIJO: Union of Agricultural Cooperatives” on the island of São Jorge (the main producer of PDO cheese). One milk sample and one sample of the NWS were taken from the respective tank in each sampling period. In addition, two samples of the curd were taken each time from different vats, so a total of 8 samples were taken. The milk samples (ca. 100 mL), the NWS (ca. 100 mL) and the cheese curd (ca. 250 g) were kept in sterile individual bottles, refrigerated (4 °C) during transport and stored at −20 °C until DNA extraction. From each of the four production dates, several batches of cheeses (from different vats) matured for 9 months were subjected to sensory analysis by the São Jorge Private Control Body and Cheese Certification. After classification, 8 cheeses granted PDO status and 8 cheeses without PDO certification were collected for analysis. The cheese samples (ca. 500 g) were vacuum packed and kept refrigerated (4 °C) until DNA extraction.

### 2.2. Sample Preparation and DNA Extraction

Bacterial cells in milk and NWS samples were concentrated by centrifugation (10 mL) at 7000× *g* for 10 min (Beckman J2-HS centrifuge). The supernatant was discarded, and the pellet was washed twice with TE buffer (Tris-EDTA: 2M Tris HCl + 0.5M EDTA, pH 8.0); the pellet was then resuspended in 1 mL of TE buffer before DNA extraction. For the preparation of cheese and curd aliquots, 5 g of the sample was weighed and 45 mL of 2% sodium citrate buffer was added, followed by homogenization in a stomacher (400 Circulator, Seward Medical, London, UK) for 2 min at 230 rpm.

Total genomic DNA was extracted using the UltraClean^®^ extraction kit Microbial DNA Isolation Kit (MoBio, Carlsbad, CA, USA). The quantity and quality of extracted DNA were evaluated by measuring absorbance at 260 and 280 nm (LVis Plate, Fluorstar Omega, BMG Labtech). The quality of the extracted DNA was confirmed via 1.5% agarose (*w*/*v*) gel electrophoresis. Only two milk samples yielded good-quality DNA after extraction. Therefore, a total of 30 samples, including raw milk (*n* = 2), NWS (*n* = 4), curd (*n* = 8), PDO cheese (*n* = 8), and non-PDO cheese (*n* = 8) were analyzed.

### 2.3. High-Throughput Sequencing

The samples were prepared for Illumina Sequencing by 16S rRNA gene amplification of the bacterial community. The DNA was amplified for the hypervariable V3–V4 region with specific primers and further reamplified in a limited-cycle PCR reaction to add sequencing adapters and dual indexes. PCR reactions were first performed for each sample using the KAPA HiFi HotStart PCR Kit according to the manufacturer’s recommendations: 0.3 μM of each PCR primer: forward primer Bakt_341F 5′–CCTACGGGNGGCWGCAG-3′ and reverse primer Bakt_805R 5′–GACTACHVGGGTATCTAATCC-3′ (Herlemann et al., 2011, Klindworth et al., 2013), and 12.5 ng of template DNA was collected accordingly in a total volume of 25 μL. PCR conditions included denaturation at 95 °C for 3 min, followed by 25 cycles of 98 °C for 20 s, 55 °C for 30 s, and 72 °C for 30 s and a final extension at 72 °C for 5 min. In the second PCR reactions, indexes and sequencing adapters were added to both ends of the amplified target region according to the manufacturer’s recommendations (Illumina, 2013). Negative PCR controls were used for all amplification procedures. PCR products were then purified in one step, normalized using the SequalPrep 96-well plate kit (ThermoFisher Scientific, Waltham, MA, USA) (Comeau et al., 2017), pooled, and pair-end sequenced in the Illumina MiSeq^®^ sequencer using V3 chemistry, according to the manufacturer’s instructions (Illumina, San Diego, CA, USA) at Genoinseq (Cantanhede, Portugal).

### 2.4. Bioinformatics

Sequence data were processed at Genoinseq (Cantanhede, Portugal). Raw reads were extracted from an Illumina MiSeq^®^ System in the fastq format and quality-filtered using PRINSEQ v. 0.20.4 [24] to remove sequencing adapters and reads with fewer than 150 bases and trim bases with an average quality lower than Q25 in a 5-base window. The forward and reverse reads were merged by overlapping paired-end reads with AdapterRemoval v. 2.1.5 [25] using default parameters. The QIIME package v. 1.8.0 [26] was used for operational taxonomic unit (OTU) generation, taxonomic identification, sample diversity and richness index calculation. Sample IDs were assigned to the merged reads and converted to the fasta format. Chimeric merged reads were detected and removed using UCHIME [27] against the Greengenes database v. 13.8 (DeSantis et al., 2006). OTUs were selected at a 97% similarity threshold using the open reference strategy. Merged reads were pre-filtered by removing sequences with a similarity below 60% against Greengenes database v. 13.8, and the remaining merged reads were then clustered at 97% similarity against the same database. The merged reads that failed clustering in the previous step were de novo clustered into OTUs at 97% similarity. OTUs with fewer than two reads were removed from the OTU table. A representative sequence of each OTU was then selected for taxonomy assignment.

### 2.5. Statistical Analysis

Alpha diversity indices Chao1, dominance, equitability, goods coverage, observed species, Shannon and Simpson were calculated to reflect the diversity and richness of bacterial communities in the different samples. Chao1 rarefaction curves were also calculated.

The OTU profiles of each sample were normalized (total sum normalization, TSS) and compared with the Bray–Curtis distance metric. The calculated Bray–Curtis distances were later used to sort the OTU profiles using principal coordinate analysis (PCoA). A Pearson correlation network was constructed based on the relative number of readings assigned to each genus in cheeses with and without PDO status. The underlying relationships between the genera observed in the cheeses were also analyzed using Spearman correlation. All analyses were performed using the program Calypso, v. 8.84 (http://bioinfo.qimr.edu.au/calypso, accessed on 29 January 2023).

To determine which genera provided significant discrimination between cheeses with and without PDO status, a stepwise discriminant analysis was performed using Wilks’ lambda. The assumptions of normality and homogeneity of the variance–covariance matrices of each group were tested using the Kolmogorov–Smirnov test and Box’s M test, respectively. To evaluate possible differences between cheeses with and without PDO status for the main taxonomic genera, the nonparametric Wilcoxon-Mann–Whitney test (for α = 0.05) was also applied. Statistical tests were performed using IBM SPSS Statistics (v. 25, IBM Corporation, New York, NY, USA).

## 3. Results

Based on 97% similarity, a total of 1612 operational taxonomic units (OTUs) were identified (out of a total of 2,039,272 sequence reads), of which, 1580 OTUs were identified in the 30 analyzed samples of milk, NWS, curd, and cheese with and without PDO status (PDO cheese and non-PDO cheese, respectively). Only 32 OTUs (representing 0.01% of the total number of reads) could not be identified. The average value of sequence frequency per sample was 67,976 reads/sample from a minimum of 48,194 reads/sample (PDO cheese) to a maximum of 89,059 reads/sample (non-PDO cheese). Although only two milk samples produced DNA for NGS, they were included in the results to understand the microbial dynamics from milk to curd.

### 3.1. Alpha Diversity

The rarefaction curve (Appendix A) showed a tendency to flatten, indicating that bacterial communities were adequately covered in all samples analyzed. This finding was confirmed by the estimated coverage index of the samples (Good’s coverage), above 99% in all samples, indicating a good description of microbial diversity (Table 1).

The richness and diversity of the bacterial community were assessed for the samples of raw milk, NWS, curd and ripened cheese with or without PDO status, and the assignment was performed using different alpha diversity indices (OTU, Chao1, dominance, equitability, Shannon index and Simpson index) as shown in Table 1. The Chao analysis, which estimates species richness, showed good richness in the samples. There were no significant differences in the Chao1 index (*p* > 0.05), in contrast to the other diversity indices (*p* < 0.05) between the species richness of milk and that of NWS, curd and ripened cheese (PDO and non-PDO). The dominance index showed a significantly higher value (*p* < 0.05) for NWS and curd than for milk and cheese (Table 1). This value indicates a several-fold lower diversity in NWS and curd, which is confirmed by the significantly lower values (*p* < 0.05) in these samples when considering the number of observed different OTUs, equitability, Shannon index and Simpson index. Conversely, greater species diversity was observed in milk, as indicated by the higher number of different OTUs (*p* < 0.05) and Shannon index (*p* < 0.05) compared to all other samples.

In contrast, an increase in species diversity, reflected in Shannon and Simpson indices, was observed from NWS to cheese (*p* < 0.05). There were no significant differences (*p* > 0.05) in species diversity between cheeses with and without PDO, although cheese with PDO resulted in slightly higher Shannon and Simpson index values (Table 1).

### 3.2. Taxonomic Composition of Bacterial Communities

The relative abundance of sequences identified at the family and genus level is shown in Figure 1. The major families found in milk were *Pseudomonadaceae* (14–52%), *Moraxellaceae* (21–4%), *Enterobacteriaceae* (13–27%) and *Streptococcaceae* (11–12%). In the M1 sample, the dominant genus was *Pseudomonas*, whereas the genus *Acinetobacter* was found in a greater proportion in the M2 sample. Although the milk had a lower abundance of bacteria of the genus *Streptococcus*, the M2 sample exhibited a greater abundance of this genus than the M1 milk sample (Figure 1b).

The dominant family in NWS was *Streptococcaceae*, with a relative abundance exceeding 99%. The *Streptococcaceae* family also dominated in curd (91 to 99%), although bacteria from the *Enterobacteriaceae* (1 to 7%), *Moraxellaceae* (<2%) and *Staphylococcaceae* (<1%) families were also detected in some samples (Figure 1a). Although communities from the *Listeriaceae* family were detected in milk (<0.3%), it should be noted that no OTUs from this family were found in NWS, curd or cheese samples. At the genus level, the bacterial population in NWS was dominated by the genus *Streptococcus* (69–92%), followed by the genera *Lactococcus* (8–31%) and *Lactobacillus* (0.004–0.9%). NWS samples W2 and W3 were characterized by a higher percentage of *Lactococcus* (Figure 1b); these samples were also characterized by the presence of bacteria belonging to the genus *Acetobacter* (0.2–0.3%). The genus *Streptococcus* was also dominant (65–91%) in the cheese curd (Figure 1b), followed by the genus *Lactococcus* (6–42%). The genera Acinetobacter (0.08–1.1%), Serratia (0–3.3%) and *Macrococcus* (0.005–1.4%) were detected in curd to a much lesser extent.

Regarding the microbiota in aged cheeses (9 months), several differences can be observed between samples of non-PDO (nPDO) and PDO cheeses (Figure 1). The *Streptococcaceae* family was dominant in non-PDO cheese samples, with the exception of sample 7 (nPDO7). However, there was a marked decrease in the *Streptococcaceae* family from curd (>99%) to cheese (29–73%). The *Lactobacillaceae* family was second-most abundant in non-PDO cheeses (12–34%), except in sample 7 (64%). In all non-PDO cheeses, the relative abundance of the *Leuconostocaceae* family was less than 5% (Figure 1a). Bacteria of the families *Staphylococcaceae* (2–3%, in samples 1 and 2) and *Enterococcaceae* (0.2–1%) were still detected in some non-PDO cheeses. In contrast, the predominant families in PDO cheeses were: *Lactobacillaceae* (26–45%), *Streptococcaceae* (24–36%), *Leuconostocaceae* (13–31%) and *Enterococcaceae* (1.5–3%). Thus, according to the sensory evaluation by the trained tasters, there was a clear difference between the cheeses that obtained PDO status and those that did not. The most striking difference concerned the relative abundance of the *Leuconostocaceae* family, which was higher in all PDO cheeses (Figure 1a). In addition, the *Lactobacillaceae* family was more represented in the PDO cheeses, whereas the *Streptococcaceae* family dominated in the non-PDO cheeses.

At the genus level, *Streptococcus* (14–58%), *Lactobacillus* (11–50%) and *Lactococcus* (9–39%) were the dominant genera of non-PDO (nPDO) cheeses. Among the subdominant microbiota, the following genera were detected: *Leuconostoc* (0.23–4.1%), *Enterococcus* (0.22–1.2%), *Staphylococcus* (0.01–1.75%), *Pediococcus* (0–1.75%), *Macrococcus* (0–1.6%), *Acinetobacter* (0–0.6%), *Weissella* (0–0.5%), *Citrobacter* (0–0.4%), *Chryseobacterium* (0–0.16%), *Delftia* (0–0.12%) and *Enhydrobacter* (0–0.11%).

In PDO cheeses, the diversity of dominant genera increased, with the genus *Lactobacillus* standing out. In these cheeses, the dominant genera were *Lactobacillus* (25–55%), *Streptococcus* (9–27%), *Leuconostoc* (8–28%), *Lactococcus* (8–26%) and *Enterococcus* (1.5–3.3%), whereas the subdominant genera were *Weissella* (0.01–2.6%), *Macrococcus* (0.04–0.75%), *Pediococcus* (0.02–0.64%), *Staphylococcus* (0.04–0.56%), *Chryseobacterium* (0.02–0.25%), *Vibrio* (0–0.23%), *Delftia* (0.04–0.17%) and *Acinetobacter* (0.01–0.16%).

Although the genus *Lactobacillus* was recently reclassified into 25 genera [28], the name of this genus is retained in this study to denote all organisms classified by 2020.

### 3.3. Beta Diversity of Bacterial Communities

The bacterial communities in the cheese, curd, NWS and milk used in cheese production differ significantly from each other, as shown by the principal coordinate analysis (PCoA, Figure 2). The first two PCoA axes accounted for 94% of the total variability, with PCoA1 and PCoA2 describing 63% and 31% of the variability, respectively. The first axis (PCoA1) refers to the differentiation of the NWS, curd and cheese populations. PCoA2 differentiates the bacterial community in milk. At both levels (family and genus), there was a high degree of dissimilarity between the bacterial community in the milk and the remaining samples. On the other hand, no differences were found between the NWS and curd samples, as they were grouped together. Some degree of dissimilarity was also found between the bacterial communities of the non-PDO and PDO cheeses, especially at the family level (Figure 2). At the genus level, one sample of cheese without PDO status (sample 7) was included in the PDO group (Figure 2b).

Cluster analysis confirmed the differentiation observed between the samples at the genus level (Figure 3). A clear separation of the milk cluster—with the highest degree of dissimilarity—from the other clusters was evident. A cluster of NWS and curd samples showed a high degree of similarity and was dominated by the genus *Streptococcus*. The cluster for non-PDO cheese included six of the eight non-PDO cheese samples and shared the high relative abundance of *Streptococcus* with the cluster for NWS and curd. On the other hand, the genera *Lactobacillus*, *Leuconostoc* and *Enterococcus* were positively differentiated in the cluster for PDO cheeses. Two samples of non-PDO cheeses (samples 6 and 7) were also included in this cluster. Although they were included in the same cluster as the PDO cheeses, these samples differed in the low abundance of OTUs of the genus *Leuconostoc* (Figure 1b).

### 3.4. Distinction of Bacterial Communities in PDO and Non-PDO Cheeses

PCoA based on Bray–Curtis distance matrix on cheese samples was performed to visualize the differences in the bacterial community between the non-PDO and PDO cheeses. As shown in Figure 4a, the bacterial communities in the PDO cheeses were closer, and more similar to each other than in the non-PDO cheeses. The results of the PCoA analysis were consistent with the network for the bacterial communities of the São Jorge cheeses (Figure 4b). The network with the interactions of OTUs identified at the genus level unfolded the difference between the PDO cheeses (blue circles) and the non-PDO cheeses (red circles). The genera *Leuconostoc*, *Enterococcus*, *Lactobacillus*, *Weissella* and *Vibrio* were associated with PDO cheeses, whereas *Streptococcus*, *Lactococcus*, *Staphylococcus*, *Citrobacter*, *Serratia*, *Enhydrobacter* and *Acinetobacter* were associated with non-PDO cheeses.

To determine which genus best characterizes PDO cheese, a stepwise discriminant analysis was performed that identified the genus *Leuconostoc* as the variable that significantly differentiates PDO cheese (*p* < 0.05). These results were confirmed by a nonparametric analysis of the OTUs assigned to the dominant genera in these cheeses (Figure 5). Compared to the non-PDO cheeses, the cheeses bearing the PDO label had a higher proportion of OTUs of the genera *Lactobacillus* (*p* < 0.05), *Leuconostoc* (*p* < 0.001), and *Enterococcus* (*p* < 0.01). In contrast, the PDO cheeses had lower *Streptococcus* OTUs (*p* < 0.05) than non-PDO cheeses.

The pattern of co-occurrence and exclusion of OTUs in the cheese samples is shown in Figure 6. A strong negative correlation is observed between *Streptococcus* and *Lactobacillus*, suggesting that a reduction in *Streptococcus* dominance is necessary for the development of *Lactobacillus* during cheese ripening. Negative correlations are also observed between *Streptococcus* and *Leuconostoc* and between *Streptococcus* and *Enterococcus*, although to a lesser extent. Conversely, the genera *Staphylococcus* and *Acinetobacter* exhibited a strong positive correlation with *Streptococcus*.

## 4. Discussion

Despite the small sample size, the results of α-diversity in milk are consistent with other studies that have found higher species diversity in raw milk compared to cheeses produced from it [7,17,29,30]. The high level of species diversity in milk decreases significantly when moving to NWS and curd. These samples have high dominance values associated with low equitability and lower Shannon and Simpson indices, indicating low diversity in bacterial community composition with dominant populations. NWSs were generally characterized by a relatively simple microbiota. This LAB community is generally thermophilic and well adapted to the particular physicochemical conditions of NWSs [31]. The decrease in biodiversity observed during the transition from milk to curd is expected because the lactic acid production of LAB from NWS lowers the pH, which contributes to cheese curd formation and inhibits pathogen growth from raw milk [32]. However, the biological richness of raw milk is of great importance as it can provide a desirable microbiota associated with specific characteristics of raw milk cheeses [32,33]. Similar results were obtained with Poro cheese, an artisanal Mexican cheese also produced from raw cow’s milk and inoculated with fermented NWS from the previous batch [29]. During the production of this cheese, the bacterial diversity in the milk was high and decreased significantly in the NWS and curd, although it increased again during cheese ripening [29].

Concerning the taxonomic composition of bacterial communities, the results were similar to those reported for milk and curd in the production of traditional Italian cheeses [34]. Other studies provided similar results to our work, with the phylum *Proteobacteria* predominant in milk, followed by *Firmicutes* and *Bacteroidetes* [17]. In contrast, Quigley et al. [35] reported that *Firmicutes* accounted for ca. 80% of the bacterial community in raw milk in Ireland. The presence of *Proteobacteria* may unfold hygiene problems in milk, as this phylum includes a wide range of Gram-negative pathogenic bacteria [36]. It should be noted that milk samples were collected from the cold storage tank, knowing that during storage, populations of psychotropic bacteria dominate, which have been reported to contribute to the spoilage of dairy products [32,37].

The present study also confirms the previous data of Kongo et al. [21], according to which *Enterobacteriaceae* were detected in the milk used for the production of São Jorge cheese. The presence of high numbers of these bacteria is generally considered an indicator of poor hygiene, and if pathogenic species are also present, this can pose a health risk; it also has a negative effect on the sensory quality of the finished cheese [35]. In contrast, the presence of the genera *Lactococcus*, *Lactobacillus*, *Leuconostoc* and *Enterococcus* in the milk samples, albeit at relatively low levels, may be critical to the development of desired flavor characteristics during cheese ripening [35]. These bacteria exhibit significant lipolytic and proteolytic activities, so they strongly influence the quality of cheese produced from raw milk [38].

As for the NWS, all samples had a bacterial community dominated by *Streptococcaceae*, which accounted for 99.1% to 99.9% of the total population. Thus, there was a significant change in the bacterial community during the transition from milk to NWS. This change was predictable since NWS was mainly associated with backslopping, and this method tends to favor the bacterial community best adapted to the fermentation of milk [39]. These results are also in agreement with those of Fontina PDO cheese, where a low correlation was found between the microbiota of raw milk and curd, which was influenced by the composition of the NWS added as a starter culture [40].

The microbial composition at the genus level of the NWS, which was dominated exclusively by *Streptococcus* and *Lactococcus*, was similar to starter cultures traditionally used in the production of aged cheese [41]. The bacteria of these genera are known to play a crucial role in acidifying milk at the beginning of cheese making. However, the less frequent presence of the genus *Lactobacillus* distinguishes this NWS from the one used in the manufacture of other artisanal cheeses [16,18,29,42,43]. In these cheeses, the NWS showed a microbiota dominated by the genera *Lactobacillus* and *Streptococcus*, as was also the case in Silter PDO cheese [6]. This difference is probably due to the heat treatment applied in the production of these cheeses (39–54 °C), i.e., higher temperatures than those commonly used for São Jorge cheese (35–36 °C). According to some authors [12,42,44], an increase in temperature during heat treatment leads to a decrease in *Lactococcus* spp. and an increase in *Lactobacillus* spp.

*Acetobacter* was also detected in NWS, but is not common in this habitat, although it has been described in some traditional cheeses (Jin et al., 2018). In addition, the presence of *Enterococcus* in NWS has been reported by some authors (Giannino et al., 2009, Silvetti et al., 2017). However, this genus was essentially not detected in the NWS samples tested. Similar results were obtained in starter cultures used in the production of Italian and Mexican cheeses [29,31]. Although the genera *Leuconostoc* and *Enterococcus* were not detected in the NWS, they were present in lower proportions in the curd, likely imported from the milk. Should they find the right conditions in the cheese ecosystem, such LAB genera would become dominant in the cheese microbiota.

The dominant microbiota in the NWS (*Streptococcus* and *Lactococcus*) was also found in the curd, whereas the dominant genera in the milk, namely, *Pseudomonas* and *Acinetobacter*, underwent a substantial reduction in the curd. These results are comparable to those reported in previous studies on different artisanal cheeses (Quigley et al., 2013, Aldrete-Tapia et al., 2014, De Pasquale et al., 2014). It is known that the changes in the food environment during the fermentation phase exert some selection pressure on the microbial populations present in raw milk, which ultimately favors the growth of LAB.

As mentioned earlier, several studies have been published on the microbiota of São Jorge cheese [19,21,22,45]. However, all of these studies resorted to cultivation-dependent methods and did not attempt to distinguish between cheeses with and without PDO status. Although cultivation-dependent methods are essential for isolating microorganisms characteristic of cheese, they may underestimate some microbial communities—particularly species that are less well-adapted to growth under conditions commonly used for isolation in the laboratory. With the recent development of new sequencing techniques, it has become possible to assess the composition of bacterial communities in these ecosystems without the bias that their isolation represents. This is, in fact, the first study to apply these methods to gain a better understanding of the microbial community of São Jorge cheese. However, this technique is limited to the identification of bacterial communities at the genus level. In addition, NSG methodologies may also introduce some bias due to the methods used in sampling, DNA extraction, PCR amplification, and sequencing (reviewed by Hugerth and Andersson [46]).

According to our results, the ripening of São Jorge cheese is dominated by the genera *Lactobacillus*, *Streptococcus*, *Leuconostoc*, and *Lactococcus*. In general, these dominant genera are similar to those previously found in ripened cheeses produced from raw milk [10,13,15,17,29,47,48,49,50,51]. Previous studies on the microbiota of São Jorge cheese also refer to *Lactobacillus* as the dominant genus at the end of ripening [19]; however, the genus *Enterococcus* accounted for 62% of isolates in the curd and 30–37% in the cheese. Given the high selection of *Enterococcus* by culture media commonly used for bacterial isolation [23], it is possible that the dominance of *Enterococcus* reported for this and other traditional cheeses was overestimated. In studies using culture-independent methods, this genus was not found expressively in the microbiota of ripening cheeses [10,17,29].

Among the lactobacilli, two species were described as dominant in an earlier study on São Jorge cheese: *Lacticaseibacillus paracasei* and *Lacticaseibacillus rhamnosus* [19]. In this study, *Lactococcus lactis* was also identified as dominant in the curd, but no *Streptococcus* spp. were isolated from the São Jorge cheese, despite the dominance of this genus observed in the present study.

As for the subdominant microbiota, the genera *Weissella*, *Macrococcus* and *Pediococcus* should be highlighted as potential contributors to cheese texture and flavor [52]. The presence of the genus *Vibrio* has also been described in Herve PDO cheese, suggesting that these bacteria may play an important role in the ripening process [53]. However, due to their low abundance and sporadic occurrence, this genus is not expected to have a positive impact on the flavor of São Jorge cheese.

To assess which genera best distinguished PDO cheeses, a discriminant analysis pointed to the genus *Leuconostoc* (*p* < 0.05). This result is not surprising since some species of the genus *Leuconostoc* are well-adapted to the cheese environment and may play an important role in flavor development during ripening [54,55]. Therefore, our results support a clear distinction between PDO and non-PDO cheeses in terms of the bacterial community. It should be noted that this classification depends solely on the evaluation of a group of tasting experts who grant (or do not) the PDO label based on the sensory characteristics of the cheese. Because this cheese is produced from raw milk without the addition of a well-defined starter culture, it is subject to wide batch-to-batch variations that often disqualify it for PDO status. When samples were taken for this work, the rejection of PDO status was over 50% of batches. Therefore, it seems crucial to know what is expected in terms of the microbiota of said PDO cheese in order to improve the sensory quality of the final product, which could eventually allow a higher percentage of PDO approval. As *Leuconostoc* has been found to be essential for the differentiation of PDO cheese, it is important to determine the factors that allow the development of these bacteria in the cheese during ripening. The differentiation resulting from the development of *Leuconostoc* may result from the environment created by the particular microbial ecology of each vat. The presence of a specific microbial community can favor the development of beneficial bacteria for the flavor development of the cheese, which guarantees the awarding of PDO status.

It should also be noted that the genera characteristic of São Jorge cheese, such as *Leuconostoc* and *Lactobacillus*, showed negative correlations with bacteria considered contaminants, e.g., *Staphylococcus* and the proteobacteria *Acinetobacter*, *Serratia*, *Klebsiella*, *Erwinia*, *Citrobacter*, *Enhydrobacter* and *Bacillus*. Similar results were reported by Zheng et al. [56], who observed a negative correlation between *Lactobacillus* and *Lactococcus*, and *Acinetobacter* and *Staphylococcus* in Kazak artisan cheese. The pattern of co-occurrence and exclusion suggests that good milk quality, low levels of contaminating bacteria and good equipment hygiene may control the dominance of *Streptococcus* during cheese ripening. Such control would allow the growth of *Leuconostoc* spp. as well as *Lactobacillus* spp. and *Enterococcus* spp., thus ensuring the proper development of the intended characteristic flavors in São Jorge PDO cheese. Thus, our results indicate that LAB populations, especially of *Leuconostoc* and *Lactobacillus*, dominate the microbiota of São Jorge PDO cheese and limit the development of spoilage bacteria, as in other cheeses [30]. Recently, *Lactobacillus* and *Lactococcus* were also shown to positively correlate with cheese quality in traditional Chinese cheeses [57].

## 5. Conclusions

The unique characteristics of São Jorge PDO cheese are related to the microbiota present in its ingredients (milk and NWS), which in turn are controlled by the production process and the ripening period. Milk stored in tanks and used for cheese production is dominated by Gram-negative bacteria of the genera *Pseudomonas* and *Acinetobacter*, whereas *Lactococcus* and *Streptococcus* were detected in lower numbers. On the other hand, the microbial composition of NWS was dominated by *Streptococcus*, followed by *Lactococcus*, which should play a positive role in curd acidification. These genera were retained in the curd, with a decrease in *Streptococcus* and an increase in *Lactococcus*. However, during ripening, a decrease in *Streptococcus* and an increase in *Lactobacillus* and *Leuconostoc* communities were observed. Thus, the microbiota of São Jorge cheese was dominated by the genera *Lactobacillus*, *Streptococcus*, *Leuconostoc* and *Lactococcus*.

This work contributed to clearly distinguishing between PDO and non-PDO cheeses in terms of bacterial community composition. PDO status is assigned using empirical methods based on sensory analysis by a tasting panel. PDO cheeses have been found to own a distinctive bacterial community in which the genus *Leuconostoc* is a distinguishing feature. *Leuconostoc* bacteria are associated with the development of flavor during the ripening process, so they should play a major role in the final sensory characteristics of São Jorge PDO cheese. In addition to the genus *Leuconostoc*, PDO cheeses were characterized by a higher occurrence of the genera *Lactobacillus* and *Enterococcus* and a lower occurrence of *Streptococcus* compared to non-PDO cheeses. The pattern of co-occurrence and exclusion of OTUs in cheese samples suggests that the presence of contaminating bacteria does not favor the development of bacteria associated with PDO status. Therefore, good milk quality appears to be essential for the development of a community rich in the genera *Leuconostoc* and *Lactobacillus* characteristic of São Jorge PDO cheese.

The results of this study will allow a better understanding of the bacterial community of São Jorge cheese and its use to distinguish between non-PDO and PDO cheeses by applying culture-independent techniques. This information is important for developing strategies to increase the percentage of cheeses that can obtain the PDO label, which will ultimately have a positive impact on the economic aspects of São Jorge cheese production.

## Figures and Tables

**Figure 1 foods-12-00990-f001:**
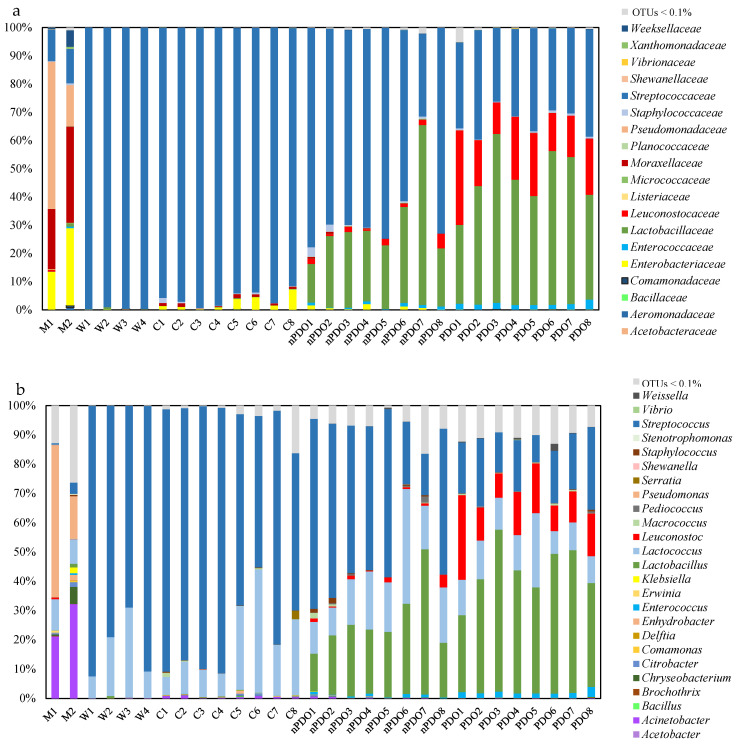
Relative abundance (%) of sequences identified at the family (**a**) and genus (**b**) level in milk (M), whey (W), curd (C), non-PDO cheeses (nPDO) and PDO cheeses (PDO). Only taxa contributing more than 0.1% of the total abundance in at least one sample are shown.

**Figure 2 foods-12-00990-f002:**
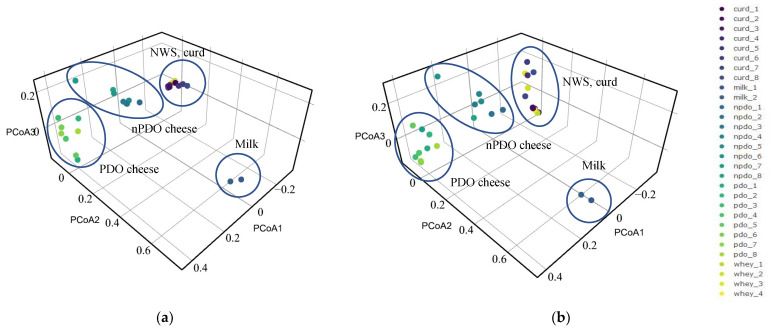
Principal coordinate analysis (PCoA) based on the Bray–Curtis distance matrix of operational taxonomic units (OTUs) identified at the family (**a**) and genus (**b**) level of milk, NWS (whey), and cheese samples (PDO and nPDO cheeses). The ellipses were drawn by hand to help visualizing the different sample types.

**Figure 3 foods-12-00990-f003:**
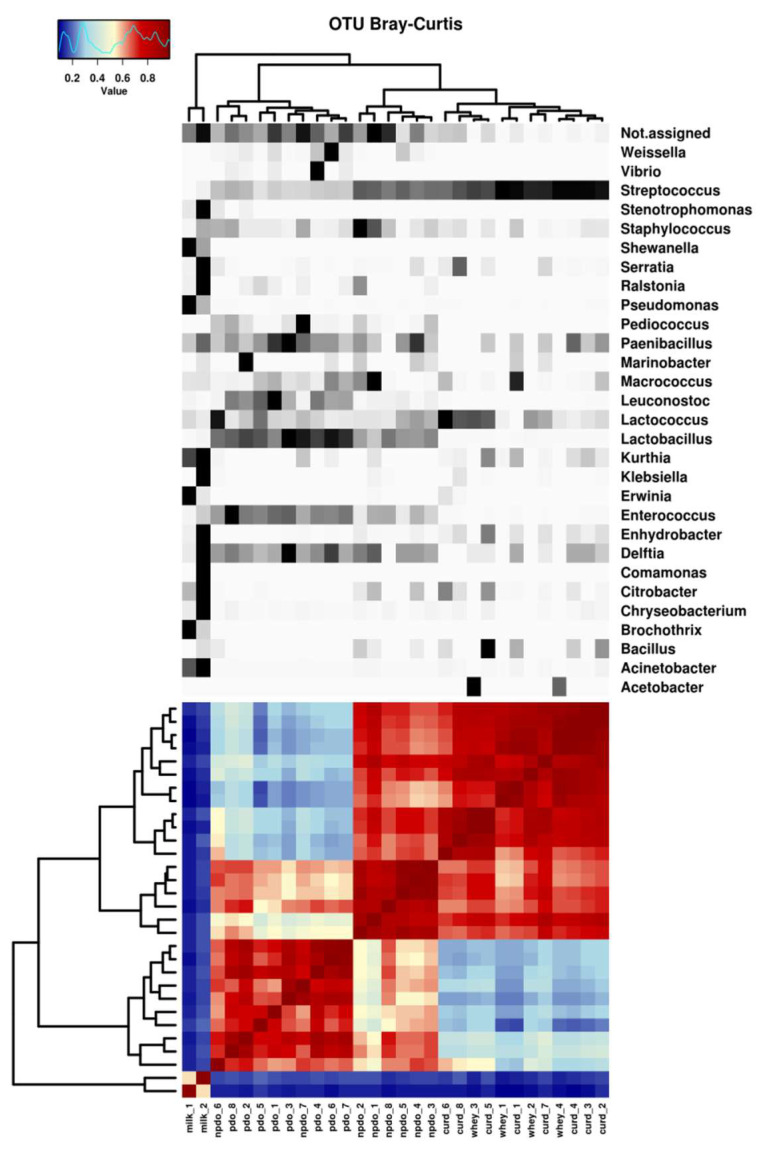
Dendrogram and heat map representing the distribution (%) of bacterial genera in samples of milk, whey (NWS), curd, non-PDO cheeses (nPDO) and PDO cheeses (PDO). Only OTUs that occur with an abundance of more than 0.1% in at least one sample were included. The grouping of the samples was obtained by hierarchical clustering. The color code, from blue to red, indicates the Bray–Curtis distance metric, where the color blue represents maximum dissimilarity and red shows maximum similarity. In the upper part, the color code indicates the relative abundance of OTUs in the sample, ranging from white (low abundance) to black (high abundance).

**Figure 4 foods-12-00990-f004:**
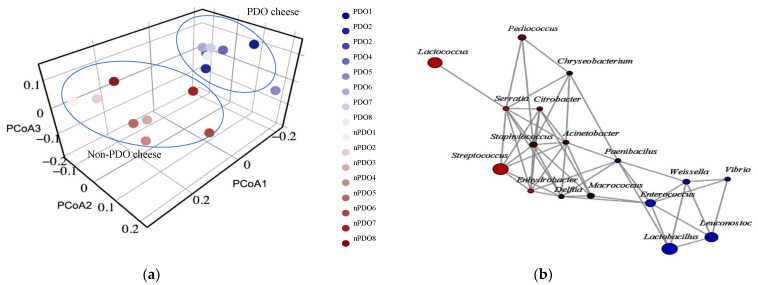
(**a**) Principal coordinate analysis (PCoA) based on the Bray–Curtis distance matrix of OTUs identified at the genus level of PDO and non-PDO cheese samples. The ellipses were drawn by hand to help visualizing the different cheese types. (**b**) Correlation network of co-occurrence patterns of OTUs classified at the genus level between non-PDO cheeses (red circles) and PDO cheeses (blue circles). The size of each node indicates the richness of OTUs belonging to each taxonomic group. Lines connecting two nodes represent significant positive correlations (*p* < 0.05).

**Figure 5 foods-12-00990-f005:**
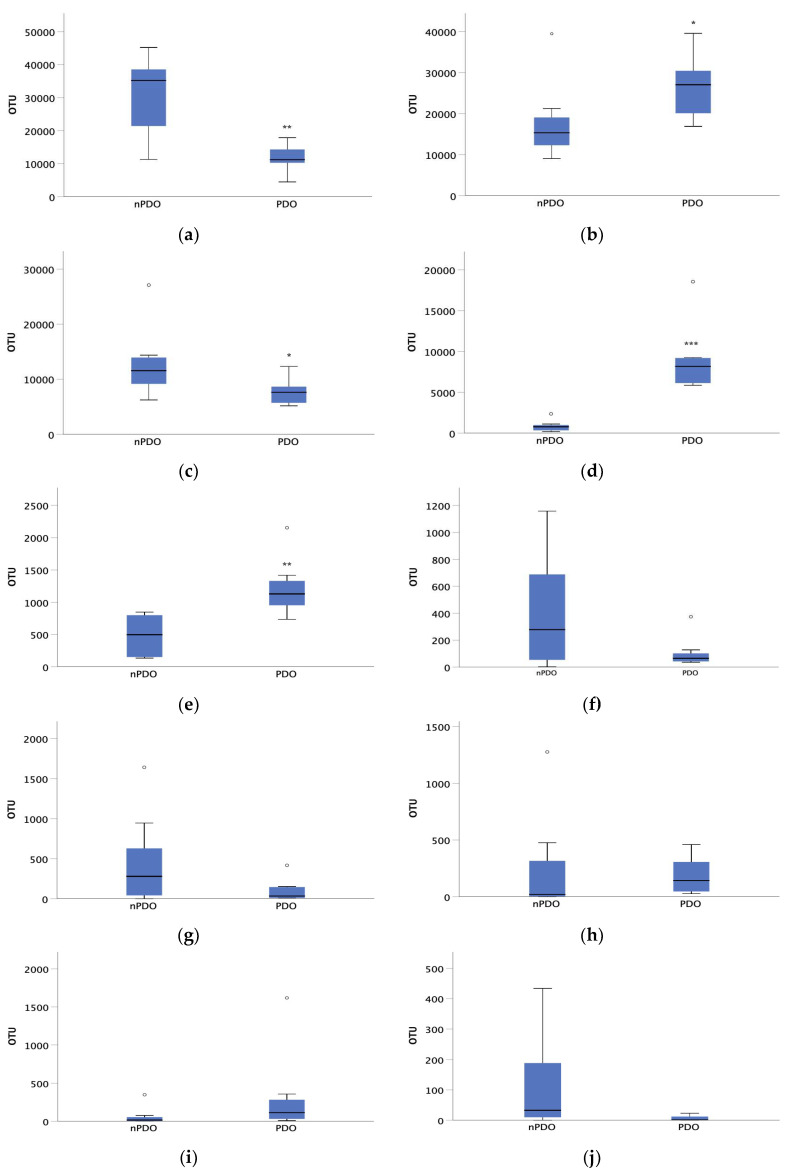
Comparison of OTUs identified at the genus level between samples of non-PDO cheese (nPDO) and PDO cheese. The genera compared were: (**a**) *Streptococcus*, (**b**) *Lactobacillus*, (**c**) *Lactococcus*, (**d**) *Leuconostoc*, (**e**) *Enterococcus*, (**f**) *Staphylococcus*, (**g**) *Pediococcus*, (**h**) *Macrococcus*, (**i**) *Weissela* and (**j**) *Serratia*. * *p* < 0.05, ** *p* < 0.01, *** *p* < 0.001.

**Figure 6 foods-12-00990-f006:**
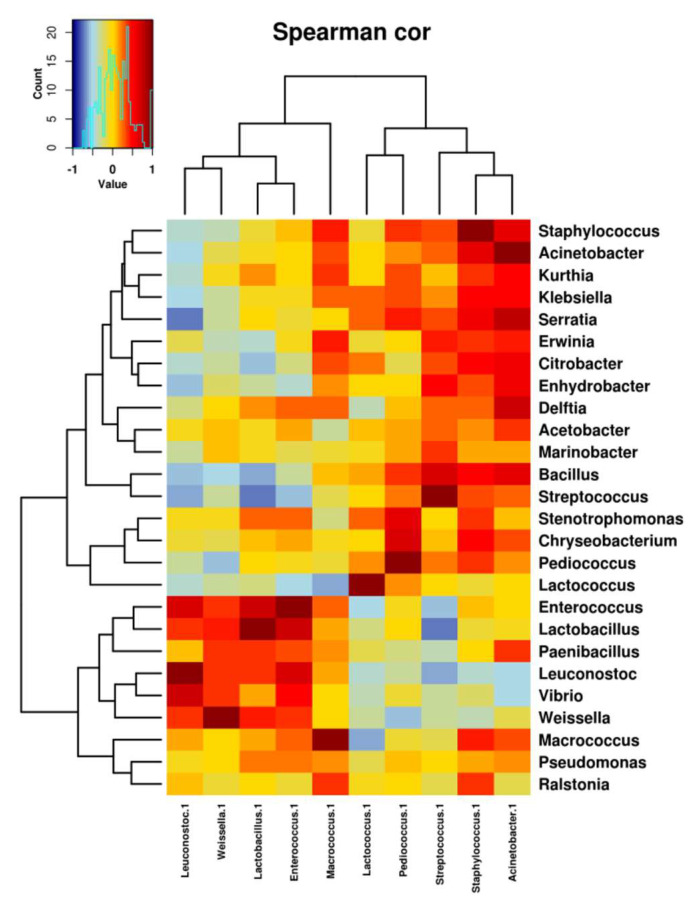
Co-occurrence and co-exclusion relationships between genera observed in cheeses, based on Spearman’s correlation. Only OTUs with a frequency greater than 0.5% in at least one sample are shown. The color of the scale bar indicates the type of correlation, with +1 indicating a positive correlation (dark red) and −1 indicating a negative correlation (dark blue).

**Table 1 foods-12-00990-t001:** Alpha diversity indices observed in samples of raw milk, NWS, curd, non-PDO cheese (nPDO) and PDO cheese (PDO). Values of the mean and standard deviation (SD) are indicated.

Samples	N	Chao1	SD	Dominance *	SD	Equitability *	SD	Good’s Coverage	SD	N. OTUs Observed *	SD	Shannon Index *	SD	Simpson Index *	SD
Milk	2	446	130	0.0935^a^	0.0697	0.5625^a^	0.0596	0.9987	0.0004	402^a^	115	4.860^a^	0.750	0.907^a^	0.070
NWS	4	302	45	0.3874^b^	0.0788	0.2467^b^	0.0441	0.9988	0.0002	139^b^	11	1.759^b^	0.340	0.613^b^	0.079
Curd	8	344	83	0.3588^b^	0.0910	0.2609^b^	0.0621	0.9986	0.0003	206^b.c^	43	2.002^b^	0.491	0.641^b^	0.091
nPDO	8	420	117	0.2000^a^	0.0267	0.3750^c^	0.0227	0.9986	0.0003	250^c^	58	2.975^c^	0.273	0.800^a^	0.027
PDO	8	401	71	0.1659^a^	0.0388	0.4084^c^	0.0235	0.9985	0.0003	232^b.c^	42	3.200^c^	0.199	0.834^a^	0.039

* Different letters within a column represent significant differences (*p* < 0.05).

## Data Availability

The data presented in this study are openly available in the Sequence Read Archive database (NCBI) under BioProject PRJNA908105.

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
