# Peer review of "Distinct Bacterial Communities in São Jorge Cheese with Protected Designation of Origin (PDO)"

_foods, 2023, doi:10.3390/foods12050990_

Round 1
Reviewer 1 Report
São Jorge cheese is produced according to Protected Designation of Origin (PDO) specifications, and the granting of the PDO label depends crucially on sensory evaluation. In the manuscript entitled “Distinct bacterial communities in São Jorge cheese with Protected Designation of Origin (PDO)”, the authors described the bacterial diversity of PDO and no-PDO cheese, and identify the specific microbiota that contributes to its uniqueness as a PDO. The study will be of strong interesting to readers. I just have several minor comments:
1. São Jorge cheese is produced from raw milk, however, the results showed that diversity of microorganism of the milk do not affect the microbial communities of the cheese, my question is the samples of milk is too small.
2. In Fig. 1, some colors are too similar (e.g in a, the Streptococcaceae, Staphylococcaceae, Enterococcaceae and Comamonadaceae), it is hard to distinguish.
3. Line 201, NWSNWS should be NWS
4. Line 203, Table 1, L304, NSW should be NWS?
Reviewer 2 Report
Line 107; DNA extraction
How was the milk concentrated by centrifugation?
Was the supernatant discarded and washed? please rewriting
Was a proteinase enzyme used to hydrolyze the proteins in DNA extraction?
Reviewer 3 Report
The authors examined the bacteria present in São Jorge PDO cheese in an effort to determine why its sensory characteristics vary. They found that Lactobacillus, Streptococcus, Leuconostoc, and Lactococcus predominated in the cheese, which will provide a better understanding of the microflora that should be observed in a high-quality product deserving of PDO status. I saw no changes that have to be made in the manuscript.
